

# An effective fingerprint orientation field estimation method using differential values of grayscale intensity

Ting-Wei Shen[1,*], Mao-Hsiu Hsu[2,*], Chun-Hsu Shen[3], Wen-Fang Wu[1], Yu-Chiao Lu[4] and Chia-Chun Chu[5]

[1] Department of Mechanical Engineering, National Taiwan University, Taipei, Taiwan
[2] Department of Information Management, Chihlee University of Technology, New Taipei, Taiwan
[3] Department of Electronic Engineering, Ming Chuan University, Taoyuan, Taiwan
[4] Department of Computer Science, National Yang Ming Chiao Tung University, Hsinchu, Taiwan
[5] Department of Aeronautics and Astronautics, National Cheng Kung University, Tainan, Taiwan
[*] These authors contributed equally to this work.

## ABSTRACT

Fingerprint orientation field (OF) estimation is important for basic fingerprint image processing and impacts the accuracy of fingerprint image enhancements, such as Gabor filters. In this article, we introduce an OF estimation algorithm based on differential values of grayscale intensity and examine the accuracy and reliability of the proposed algorithm by applying it to fingerprint images processed using Gaussian blurring and the Gaussian white noise process. The experimental results indicate that the OF estimation reliability of the proposed algorithm is higher than the gradient-based method and the power spectral density (PSD) based method in low quality fingerprints. The proposed algorithm is especially useful in noisy fingerprint images, where the OF estimation reliability of the algorithm is 6.46% and 32.93% higher than the gradient-based method and the PSD-based method, respectively.

Corresponding authors
Ting-Wei Shen,
wayne883113@gmail.com,
d05543008@ntu.edu.tw
Chun-Hsu Shen,
chshen0656@mail.mcu.edu.tw

## INTRODUCTION

Fingerprints are one of the most commonly used and universally recognized features in personal identification (*Abate et al., 2007*; *Maltoni et al., 2009*; *Ratha et al., 1996*; *Jain, Ross & Prabhakar, 2004*). There are two main reasons fingerprints are so commonly used: first, fingerprints are unique and immutable for each person, so they are ideal indicators of an individual's identity (*Alqadi et al., 2020*; *Raja, 2010*); second, fingerprints have the highest reliability of all biometric indicators (*Berry & Stoney, 2001*; *Newham, Bunney & Mearns, 1995*). Fingerprint identification technologies are widely used in various fields (*Jain, Ross & Prabhakar, 2001*). For example, forensic experts often use fingerprints in criminal investigations (*Investigation, 1984*).

Fingerprint matching methods either use global or local fingerprint features (*Cappelli, 2011*). Extracting fingerprint features from imaging is highly dependent on the integrity

and quality of the fingerprint images (*McMahon et al., 1975*; *Altarawneh et al., 2007*), so fingerprint enhancements are often used to improve the quality of the fingerprint images. There are many techniques and methods used for fingerprint enhancement so that the fingerprint features contained in the images can be displayed more clearly and extracted more accurately (*Ahmed et al., 2015*; *Marques, 2011*). Current fingerprint image enhancement algorithms reduce noise and increase contrast in grayscale fingerprint images; the Gabor filter is one of the classic fingerprint image enhancement algorithms (*Yang et al., 2003*).

Orientation field (OF) is the main variable in the Gabor filter, and a reliable OF in the preprocessing step of the Gabor filter is necessary for accurate results (*Hong, Wan & Jain, 1998*; *Turroni et al., 2011*). and for choosing a suitable Gabor filter from the Gabor filter Bank. If the estimated OF is incorrect, such as if the estimated orientation is perpendicular to the actual fingerprint orientation, then the Gabor filter distorts the fingerprint image, resulting in incorrect fingerprint feature identification and extraction (*Gottschlich, Mihailescu & Munk, 2009*), so preventing a poor fingerprint OF estimation is also important for ensuring accurate results (*Dyre & Sumathi, 2017*).

In recent decades, different OF estimation methods have been proposed, including gradient-based methods (*Awad, 2016*; *Li et al., 2018*), slit-based approaches (*Oliveira & Leite, 2008*), frequency domain-based estimations (*Ciezar & Pochylski, 2022*; *Park & Park, 2005*), learning-based models (*Cao & Jain, 2018*; *Qu et al., 2018*), and gray-level variance methods (*Dyre & Sumathi, 2017*; *Bian et al., 2019*; *Turroni et al., 2011*). Currently, the most popular of these fingerprint image OF estimation approaches is the gradient-based method because of its high resolution, high accuracy, and low computational demand (*Turroni et al., 2011*; *Sharma & Dey, 2019*; *Ratha, Chen & Jain, 1995*; *Gottschlich, Mihailescu & Munk, 2009*; *Bazen & Gerez, 2002*; *Liu & Dai, 2006*; *Kekre & Bharadi, 2009*; *Mei, Sun & Xia, 2009*; *Wieclaw, 2013*; *Wang, Hu & Han, 2007*; *Bazen & Gerez, 2000*). However, the gradient-based method is sensitive to image quality and may fail to accurately estimate the OF in low-quality fingerprint images (*Wang, Hu & Han, 2007*; *Galar et al., 2015*).

To address the limitations of the gradient-based method, we propose an orientation field (OF) estimation method based on differential values of grayscale intensity. Experimental results comparing the proposed method to the classic gradient-based estimation and another commonly used OF estimation method, power spectral density (PSD) based estimation, demonstrate that the proposed method is more accurate and reliable in low-quality fingerprint images that exhibit Gaussian blurring or Gaussian white noise. The proposed method utilizes only convolution calculations to obtain the fingerprint OF estimation.

## METHODS

### Differential values of grayscale intensity in each orientation

For each fingerprint image, we drew straight lines through the center of the image in $N$ (4, 8, 16, 32, *etc.*) orientations, and plotted line graphs of the grayscale values of the image along each straight line. For example, Figure 1 shows 16 straight lines with 16

orientations being used for the fingerprint image. If the straight lines are drawn more orthogonal to the orientation of the fingerprint, the grayscale values are more sensitive along the orientation of the lines. In contrast, if the straight lines are drawn close to the orientation of fingerprint, the grayscale value is less sensitive to changes. Figure 2 shows the line graphs of the grayscale values of the 16 orientations, where the $x$-axis is the index of each pixel and the $y$-axis is the corresponding grayscale value of the pixel. We then differentiated the grayscale value of each orientation and calculated its absolute value to represent the degree of change in the grayscale value, with smaller values indicating that the orientation of the straight line is closer to the orientation of the fingerprint . Because of smaller variances in grayscale values, we estimated that the fingerprint orientation of the fingerprint image in Finger 2 is around 56.25°. The aim of this study is to create an efficient and dependable algorithm for OF estimation using the aforementioned concept. To simplify the calculations, differential operations will be substituted with convolution calculations. Figure 3 provides the calculation steps used in this proposed method. In our experimental results, the performance of the proposed OF estimation increased as the number of orientations increased. However, because the convolution kernel size should increase with the number of orientations, the computational demand of 32 orientations is much higher than 16 orientations, and the convolution kernel size should also not be larger than the pixels between fingerprint ridges (10 pixels), so we selected 16 orientations for this study.

## Image slicing

The first step of the proposed OF estimation algorithm is slicing each fingerprint image into 10 equal parts along the $x$ and $y$ axes (Cartesian coordinates), resulting in 100 blocks of 16 x 16 pixels from each fingerprint image. We decided on 10 divisions on each axis because each small block should not be smaller than the kernel size of convolution. We then estimated the orientation field for each individual block. Figure 4 depicts image slicing.

## Image convolution

The second step of the proposed algorithm is performing a convolution calculation on each block, as shown in Eq. (1), where $x$ is the grayscale value of the image, $h$ is the convolution kernel, $y$ is the convolution result, $m$ and $n$ denote the column and row of the block, and $i$ and $j$ denote the column size and row size of the kernel, respectively. Our method performs a convolution calculation on each block with 16 kernels of $5 \times 5$ each, and each kernel representing a different orientation. Figure 5 shows the calculation results of the 16 kernels. The angular difference of each direction was 11.25 degrees, which is in line with the common Gabor filter bank (*Medina et al., 2017*; *Mukherjee & Das, 2021*), and the result of the convolution calculation was related to the differential grayscale value in each orientation. For 2D function $f(x,y)$, the partial differential equation is shown in Eq. (2) and for discrete data, we can approximate using finite differences with Eq. (3). Then, we discuss about the Prewitt filter, which comprises two convolution kernels of size $3 \times 3$ each (shown in Fig. 6). These kernels are specifically designed to detect horizontal and vertical edges and can also be applied independently to determine the gradient component

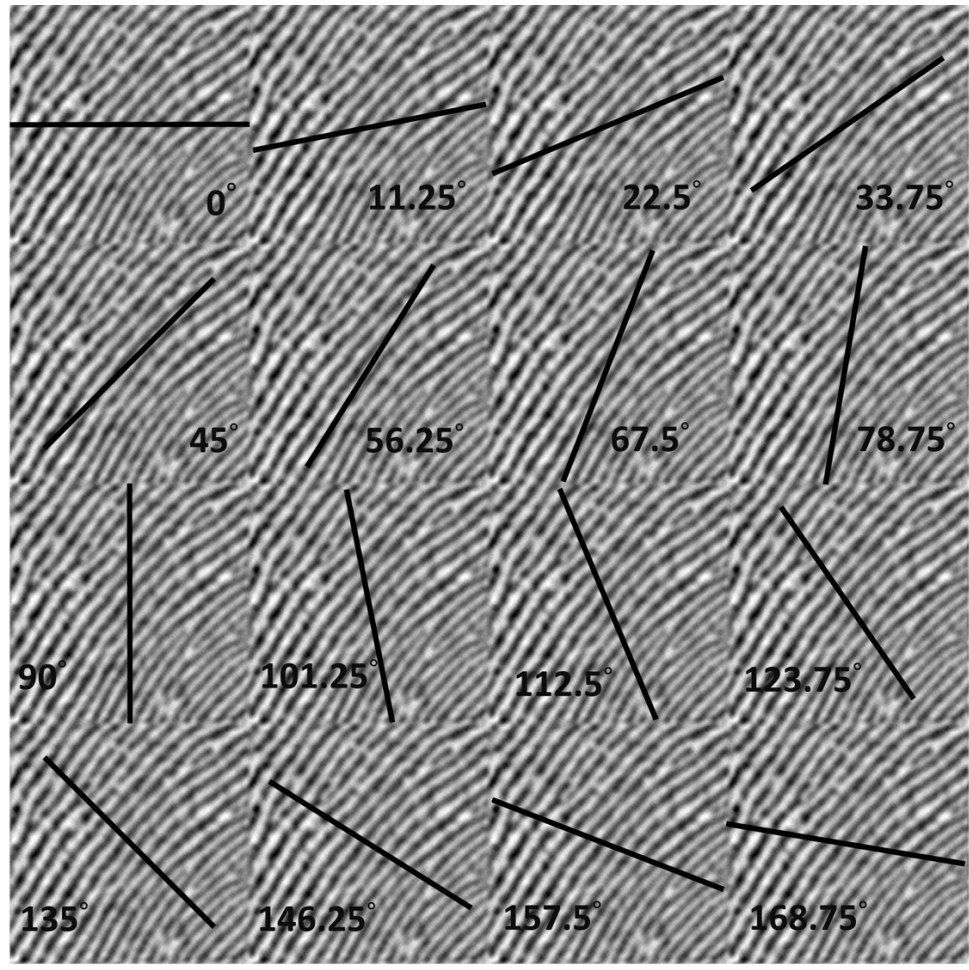

**Figure 1** Schematic view of 16 orientations of a fingerprint image.

in each orientation (as *Gx* and *Gy*), which is equivalent to the partial differential results for 2D images (Eq. (3)). Therefore, in this article, we utilize this concept to calculate the differential using convolution.

$$y[m,n] = x[m,n] * h[m,n]$$
$$= \sum_j \sum_i x[i,j] \cdot h[m-i,n-j] \tag{1}$$

$$\frac{\partial f(x,y)}{\partial x} = \lim_{\varepsilon \to 0} \frac{f(x+\varepsilon,y) - f(x,y)}{\varepsilon}$$
$$\frac{\partial f(x,y)}{\partial y} = \lim_{\varepsilon \to 0} \frac{f(x,y+\varepsilon) - f(x,y)}{\varepsilon} \tag{2}$$

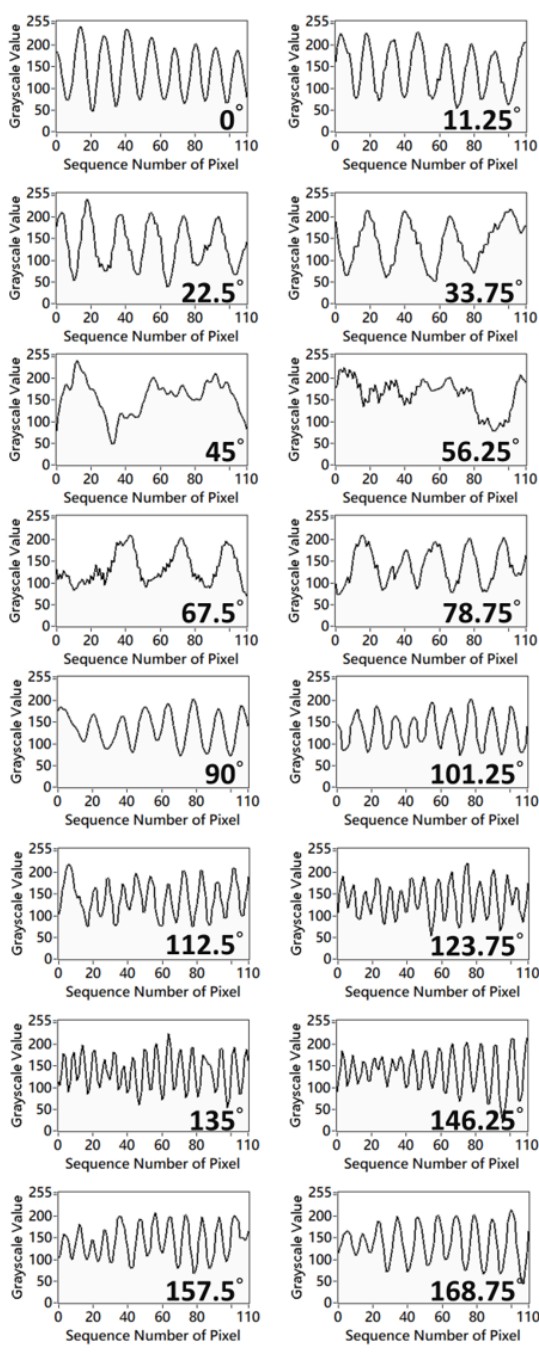

**Figure 2** Line graphs of grayscale values of straight lines in the 16 orientations shown in Fig. 1.

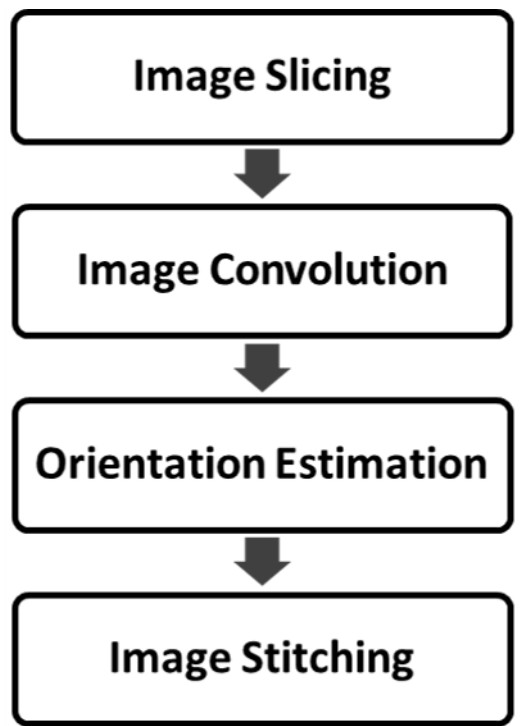

**Figure 3** The proposed OF estimation process.

$$\frac{\partial f(x,y)}{\partial x} = G_x \approx \frac{f(x+1,y)-f(x,y)}{1}$$
$$\frac{\partial f(x,y)}{\partial y} = G_y \approx \frac{f(x,y+1)-f(x,y)}{1}. \tag{3}$$

### Orientation field estimation (index, matrices)

After image convolution, the third step entails computing the sum of the absolute grayscale values for each block (represented as $Y$) for all 16 orientations (kernels). Then, the minimum $Y$ value is determined using Eq. (4), whereby a higher $Y$ value indicates a more substantial change in grayscale value in that orientation and a more orthogonal orientation relative to the fingerprint orientation. Conversely, a lower $Y$ value implies greater parallelism with the fingerprint orientation. To estimate the orientation field, the minimum two $Y$ values are weighted, and their weighted average is computed using Eq. (5).

$$Y = \sum_n \sum_m |y[m,n]|$$
$$Y_{\min} = \min(Y_1, Y_2, Y_3, \ldots, Y_{16}) \tag{4}$$

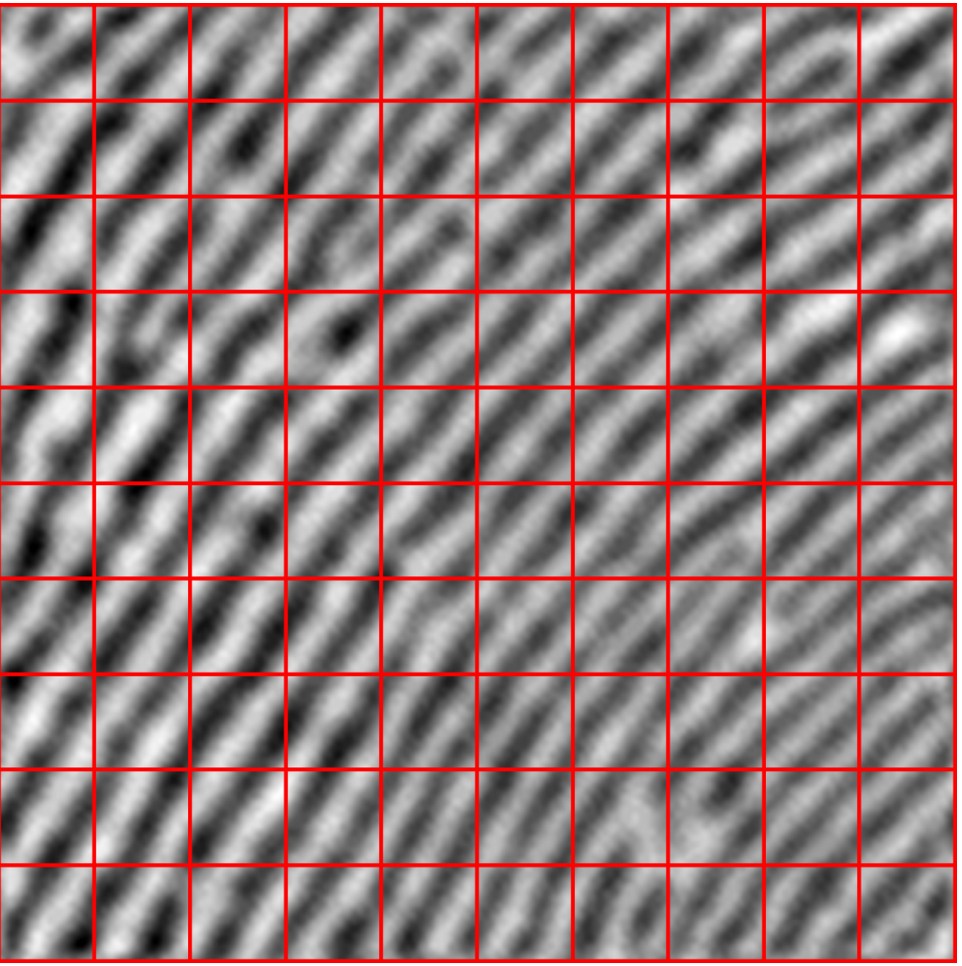

**Figure 4** **Schematic view of fingerprint image slicing.**

$$OF_{\text{weighted average}} = OF_{\min 1} \times \frac{\frac{1}{Y_{\min 1}}}{\frac{1}{Y_{\min 1}} + \frac{1}{Y_{\min 2}}} + OF_{\min 2} \times \frac{\frac{1}{Y_{\min 2}}}{\frac{1}{Y_{\min 1}} + \frac{1}{Y_{\min 2}}}. \tag{5}$$

### Image stitching

The fourth step of the proposed algorithm is stitching the 100 blocks back together to display the estimated fingerprint OF for the whole fingerprint image. Figure 7 shows the fingerprint image and resulting OF.

## RESULTS AND DISCUSSION

We designed three experiments to verify the accuracy and reliability of the proposed algorithm. In Experiment 1, we selected 15 fingerprint images (as shown in Fig. 8A), calculated the OF estimations with the proposed algorithm, and compared these results with the results of the classic OF gradient-based method and PSD-based method to

**Figure 5** Orientation convolution kernels.

determine the accuracy of the proposed OF estimation algorithm in clear fingerprint images. In Experiment 2, we performed Gaussian blurring on each fingerprint image (as shown in Fig. 8B). We calculated the OF estimations of the original images and the Gaussian blurred images using the proposed algorithm, the gradient-based method, and the PSD-based method, and compared the results of all three methods between the original and Gaussian blurred fingerprints to determine the reliability of the proposed algorithm in blurred fingerprints. In Experiment 3, we performed the Gaussian white noise process on each fingerprint image (as shown in Fig. 8C), calculated the OF estimations of the original images and the images with Gaussian white noise using the proposed algorithm, the gradient-based method, and the PSD-based method, and compared the results to determine the reliability of the proposed algorithm in noisy fingerprint images. All algorithms were developed and implemented in LabVIEW. The LabVIEW VDM (Vision

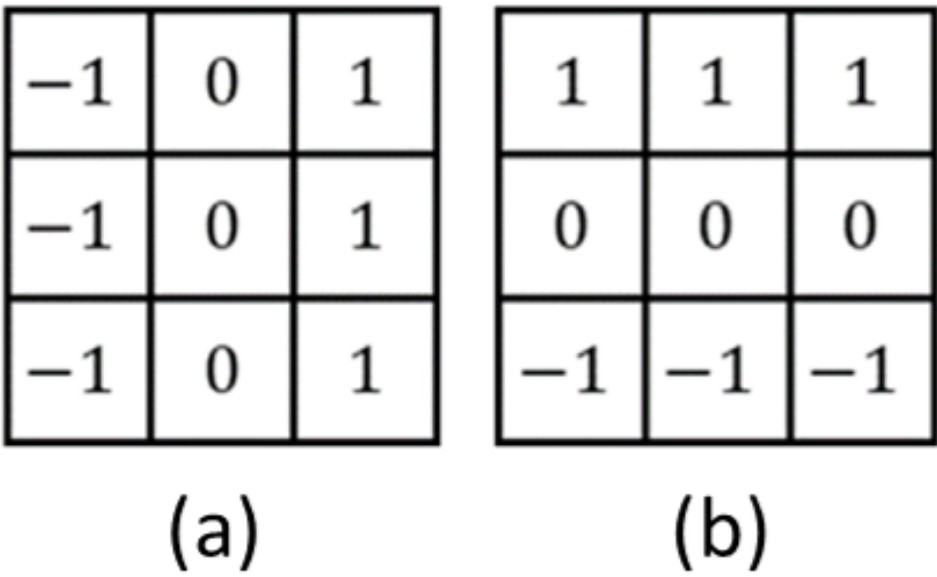

**Figure 6** **The Prewitt filter kernels.** (A) *Gx*; (B) *Gy*.

Development Module) is a powerful tool designed to develop machine vision applications and was suitable for this work.

## Database

In this study, we used the fingerprint data of volunteers, which were captured with an OLED panel display image sensor. These fingerprint images were only used for orientation field estimation, which was explained to the study participants. No personal information such as age, gender, or name, was collected from the volunteers. The final study dataset included 2520 images (126 finger samples, with 20 elements each). The resolution of each sample is 813 DPI and 160 × 160 pixels.

## Experiment 1: accuracy assessment

To validate the accuracy of our proposed orientation field (OF) estimation algorithm, we conducted an experiment with a "control group" using the commonly-used gradient-based method. We selected 15 fingerprint images from our study database and computed the OF using the proposed algorithm, the gradient-based method, and the PSD-based method. The results were then compared and recorded. Our proposed algorithm's estimated OF was found to be similar to that of the gradient-based method, whereas the PSD-based method had more differences with the gradient-based method. Figure 9 shows that the proposed algorithm had a deviation of more than 20 degrees in 67 blocks, while the PSD-based method had a deviation of more than 20 degrees in 119 blocks. We considered these deviations as misinterpretations, and the 67 blocks were divided by the total number of 1,500 blocks from the 15 fingerprint images to obtain an accuracy rate of 95.53% for our proposed algorithm, which is higher than the 92.06% accuracy rate for the PSD-based method. The error blocks

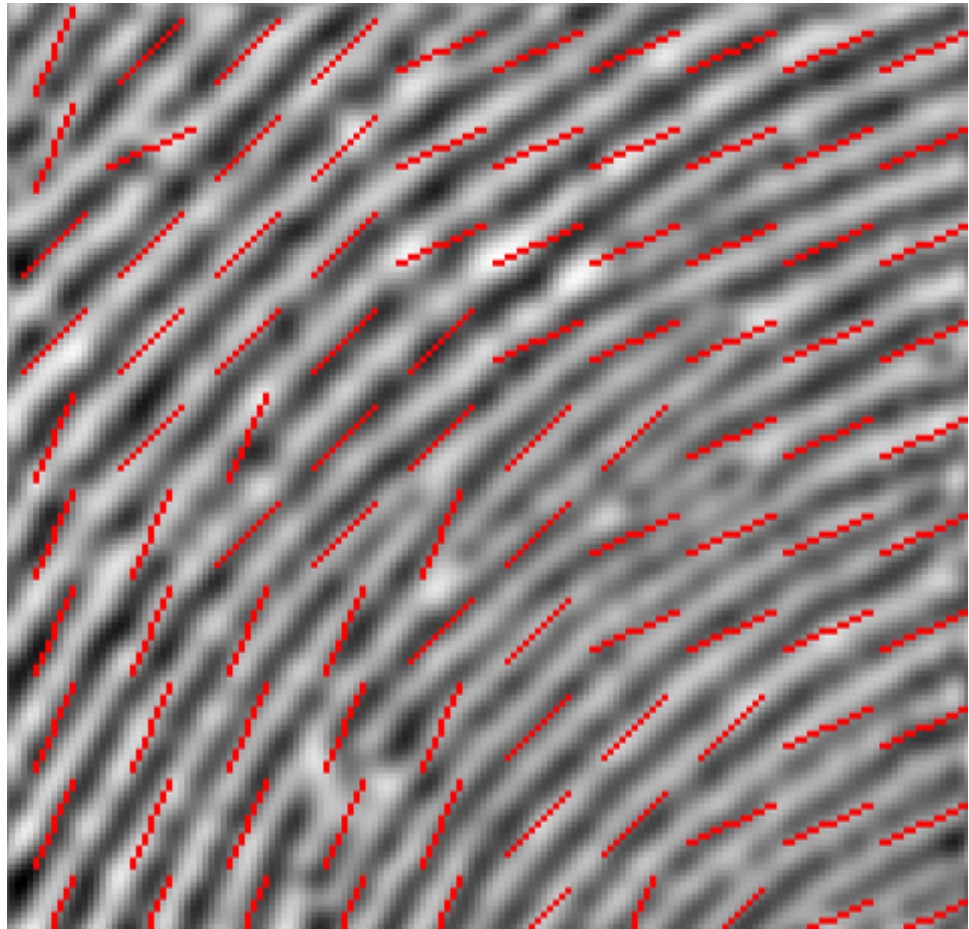

**Figure 7** Fingerprint image and fingerprint OF.

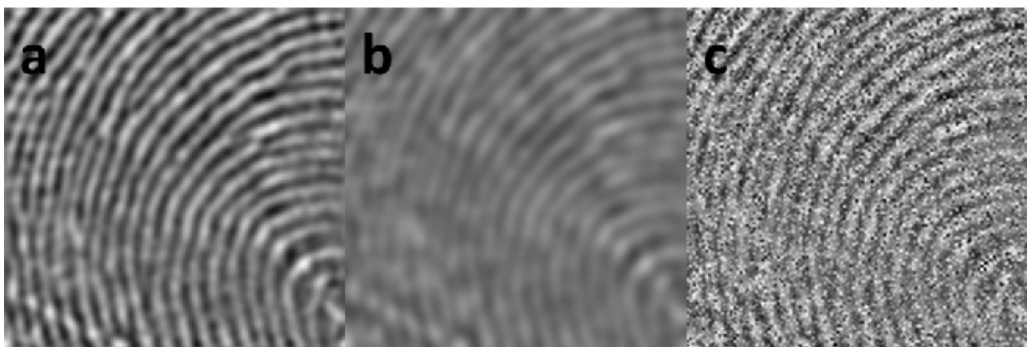

**Figure 8** Fingerprint image samples for experiments: (A) Original sample; (B) blurred sample; (C) noisy sample.

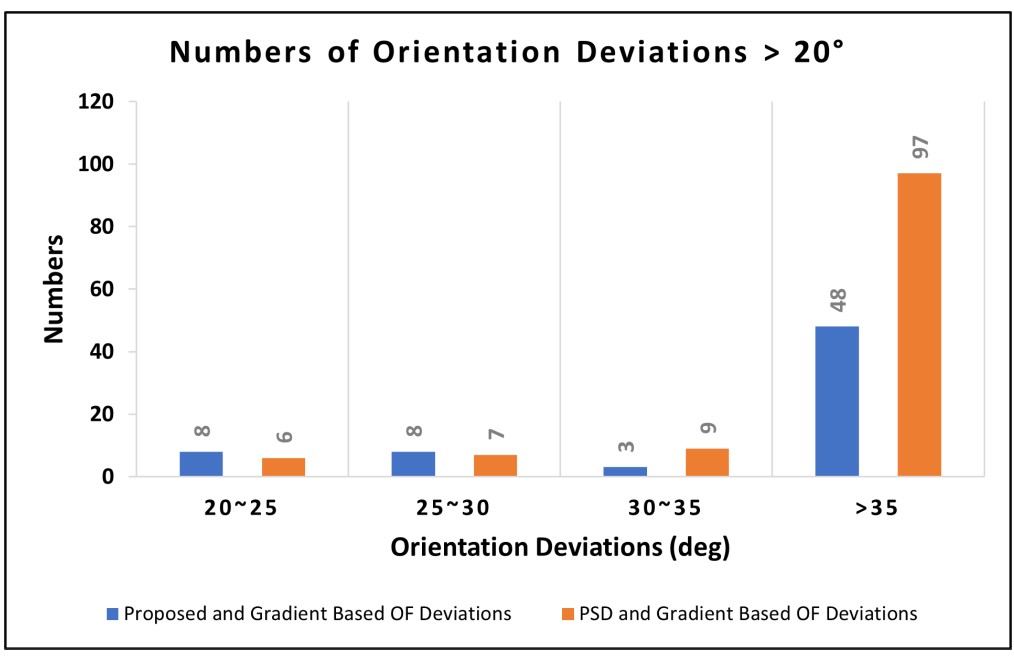

**Figure 9** Deviations in the estimated OF between the proposed method/the PSD-based method and the gradient-based method on clear fingerprint images (1,500 blocks).

in the fingerprints were primarily concentrated around the edges (as shown in Fig. 10) and there is a likelihood that error blocks were also present in the core area of the fingerprints. This could be attributed to the fact that the edges of the fingerprints tend to have the lowest image quality, which makes it more challenging to obtain accurate data. Furthermore, the core area of the fingerprint is a semi-circle or arc, which lacks obvious orientations, making it difficult to define the orientation of the overall fingerprint in those core blocks. According to the results of the correlation coefficient analysis, the proposed algorithm showed a strong positive correlation ($r = 0.909$) with the gradient-based method, indicating that the two methods are highly correlated. In comparison, the correlation coefficient between the PSD-based method and the gradient-based method was found to be 0.755, which is lower than the correlation coefficient between the proposed algorithm and the gradient-based method. These findings indicate that the proposed algorithm a more reliable and accurate method compared to the PSD-based method.

## Experiment 2: accuracy assessment in blurred fingerprint images

To verify the reliability of the proposed OF estimation algorithm on blurred images, we performed Gaussian blurring on the 15 fingerprint images selected in Experiment 1. The Gaussian blur (the Gaussian smooth) is a nonuniform, low-pass filter that blurs the details of an image, but preserves low spatial frequencies and reduces image noise. This is done by convoluting an image with a Gaussian kernel; this study used a $7 \times 7$ kernel, as shown in Fig. 11. We calculated the estimated OF of the original images and the Gaussian blurred images using the proposed algorithm, the gradient-based method, and

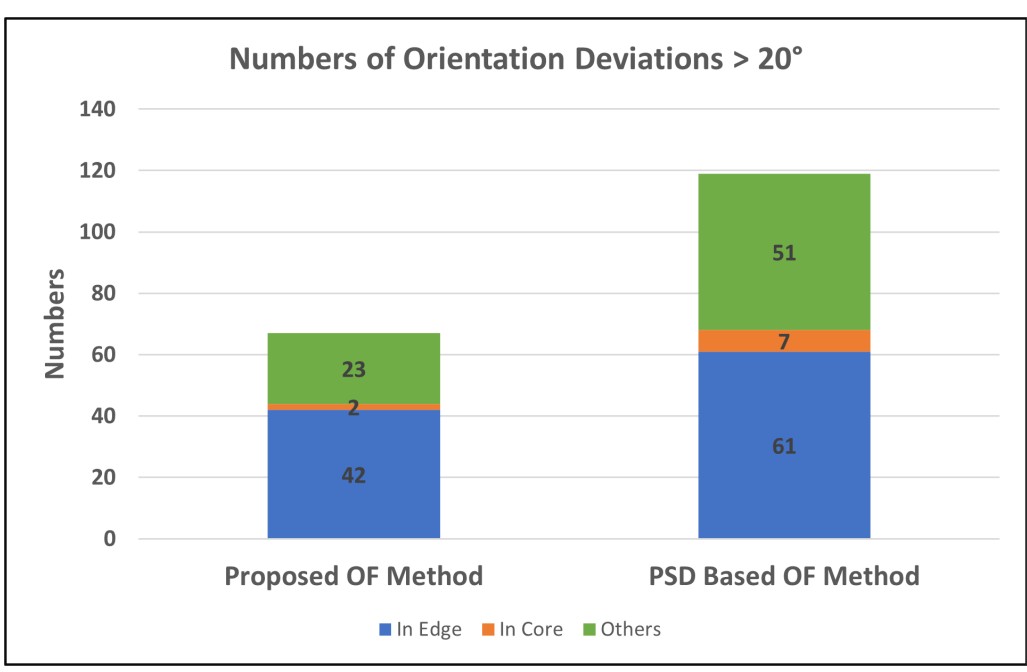

**Figure 10** Deviations in the estimated OF between the proposed method/the PSD-based method and the gradient-based method on clear fingerprint images (1,500 blocks) by location.

the PSD-based method, and compared the results of all three methods between the original and Gaussian blurred fingerprints to determine the reliability of the proposed algorithm in blurred fingerprints. The histograms of the results (shown in Fig. 12) show that there was only a small difference between the estimated OF results of the clear and blurred fingerprint images calculated by the proposed algorithm. Seven blocks had an orientation deviation of more than 20 degrees, indicating an accuracy rate of 99.53%, the average orientation deviation was 1.81°, and the correlation coefficient was 0.995, which shows that the proposed algorithm is reliable for estimating OF in blurred fingerprint images. The OF estimation results of the clear and blurred fingerprint images calculated by the gradient-based method resulted in 13 blocks with an orientation deviation of more than 20 degrees, indicating an accuracy rate of 99.13%, the average orientation deviation was 2.18°, and the correlation coefficient was 0.978. The PSD-based OF estimation results of the clear and blurred fingerprint images resulted in 126 blocks with an orientation deviation of more than 20 degrees, an accuracy rate of 91.6%, an average orientation deviation of 6.92°, and the correlation coefficient was 0.868. These results indicate that the OF estimation performance of the proposed method was slightly better than the OF estimations of the gradient-based and PSD-based methods in blurred fingerprint images.

## Experiment 3: accuracy assessment in noisy fingerprint images

To verify the reliability of the proposed OF estimation algorithm in noisy fingerprint images, we performed the Gaussian white noise process on the 15 selected fingerprint images. The Gaussian white noise process is a signal processing technique used to add random noise

| 0 | 0 | 1 | 2 | 1 | 0 | 0 |
|---|---|---|---|---|---|---|
| 0 | 3 | 13 | 22 | 13 | 3 | 0 |
| 1 | 13 | 59 | 97 | 59 | 13 | 1 |
| 2 | 22 | 97 | 159 | 97 | 22 | 2 |
| 1 | 13 | 59 | 97 | 59 | 13 | 1 |
| 0 | 3 | 13 | 22 | 13 | 3 | 0 |
| 0 | 0 | 1 | 2 | 1 | 0 | 0 |

**Figure 11  The Gaussian blurring kernel.**

values from a Gaussian distribution to the original pixel values of an image. The probability distribution function for a Gaussian distribution has a bell shape (or normal distribution). We then calculated the estimated OF of the original images and the images with Gaussian white noise using the proposed algorithm, the gradient-based method, and the PSD-based method, and compared the results to determine the reliability of the proposed algorithm in noisy fingerprint images. The experimental results (shown in Fig. 13) showed that the differences in estimated OF in the clear and noisy fingerprint images were much larger than in blurred fingerprints (Experiment 2) among all three algorithms. The proposed algorithm resulted in 383 blocks with an orientation deviation of more than 20 degrees, giving it an accuracy rate of 74.46%, the average orientation deviation was 13.86°, and the correlation coefficient was 0.898. The OF estimation results of the gradient-based method produced 480 blocks with an orientation deviation of more than 20 degrees, an accuracy rate of 68%, an average orientation deviation of 21.79°, and the correlation coefficient was 0.661. The OF estimation results of the PSD-based method produced 877 blocks with an orientation deviation of more than 20 degrees, an accuracy rate of 41.53%, an average orientation deviation of 42.19°, and the correlation coefficient was 0.236. These results indicate that the proposed method outperforms both the gradient-based and PSD-based

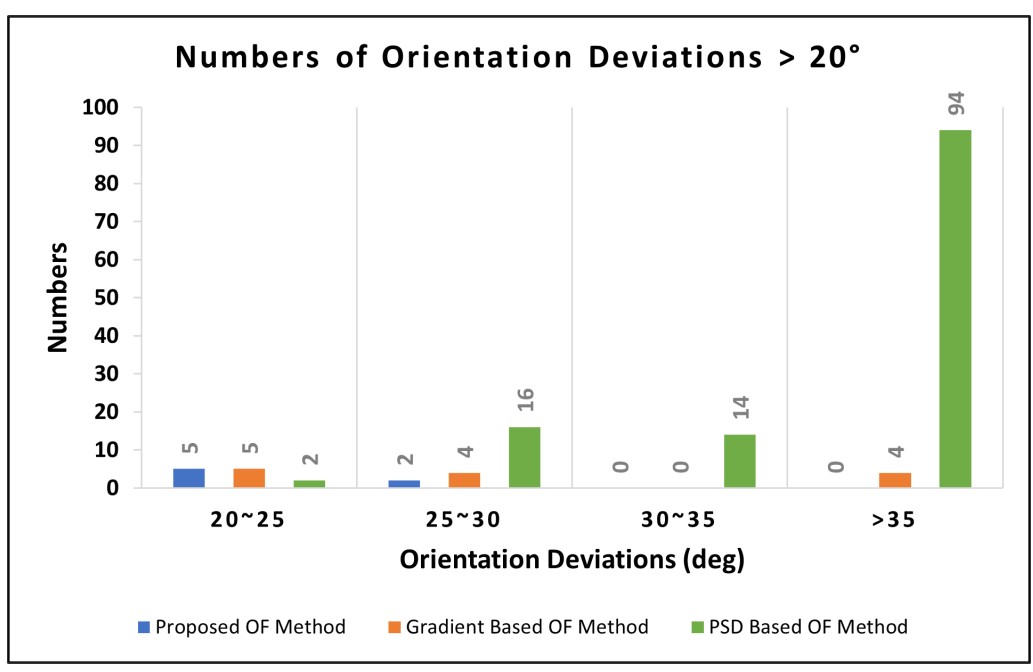

**Figure 12** Deviations in estimated OF between clear and blurred fingerprint images (1,500 blocks).

methods in OF estimation in noisy fingerprint images. A comparison of the performance and positive/negative aspects of the proposed algorithm and the two classic OF estimation methods is shown in Tables 1 and 2.

## CONCLUSIONS

Fingerprint identification technologies are widely used. Low-quality fingerprint images inhibit accurate fingerprint feature extraction, classification, and recognition, so advanced image processing techniques, like the Gabor filter, are necessary for enhancing image sharpness and reducing noise in fingerprint images. These techniques require a reliable OF estimate for accurate feature extraction.

In this study, we proposed an effective fingerprint OF estimation method based on grayscale intensity. Experimental results showed that the orientation fields estimated by the proposed OF estimation algorithm and the gradient-based method were similar, with 67 blocks with a deviation of more than 20 degrees observed between the two results, giving the proposed algorithm a 95.53% accuracy rate compared to the gradient method (from 1,500 blocks in total). In blurred fingerprint images, the OF estimation reliability of the proposed OF estimation method was 0.4% (from 99.13% to 99.53%) and 7.93% (from 91.6% to 99.53%) higher than the gradient-based method and PSD-based method, respectively. The OF estimation reliability of the proposed method in noisy fingerprint images was 6.46% (from 68% to 74.46%) and 32.93% (from 41.53% to 74.46%) higher than the gradient-based method and PSD-based method, respectively. These results indicate that the proposed algorithm is much more reliable in estimating OF in blurred and noisy

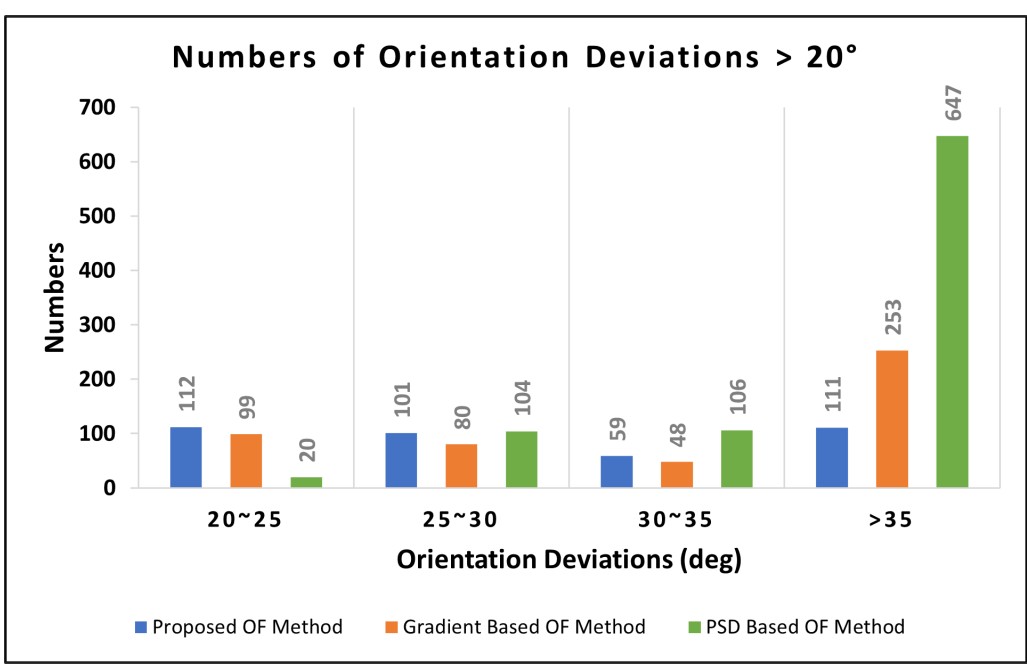

**Figure 13** Deviations in estimated OF between clear and noisy fingerprint images (1,500 blocks).

**Table 1** The performance of the proposed OF estimation method, the gradient-based method, and the PSD-based method on blurred and noisy fingerprint images.

|  | Proposed of estimation | Gradients-based of estimation | PSD-based of estimation |
|---|---|---|---|
| Numbers of deviations > 20° in blurred fingerprint images | 7 | 13 | 126 |
| Accuracy in blurred fingerprint images | 99.53% | 99.13% | 91.6% |
| Average deviations in blurred fingerprint images | 1.81° | 2.18° | 6.92° |
| Correlation coefficient between clear and blurred fingerprint images | 0.995 | 0.978 | 0.868 |
| Numbers of deviations > 20° in noisy fingerprint images | 383 | 480 | 877 |
| Accuracy in noisy fingerprint images | 74.46% | 68% | 41.53% |
| Average deviations in noisy fingerprint images | 13.86° | 21.79° | 42.19° |
| Correlation coefficient between clear and noisy fingerprint images | 0.868 | 0.661 | 0.236 |

**Table 2** Advantages and disadvantages of the proposed method and the classic OF estimation methods.

|  | Proposed of estimation method | Gradients-based of estimation method | PSD-based of estimation method |
|---|---|---|---|
| Advantages | Much higher reliability in blurred and noisy fingerprint images | High accuracy and resolution in clear fingerprint images | Lower computational time than the proposed algorithm. |
| Disadvantages | Longer computational time demand | Lower reliability in blurred and noisy fingerprint images | Worst reliability in blurred and noisy fingerprint images |

fingerprint images than the two commonly used methods tested. This may be because the gradient-based method first calculates the gradient and then uses orthogonality and gradient $Gx/Gy$ division to calculate OF, so it is greatly affected by noise. Conversely, the proposed OF algorithm uses convolution, which uses only addition and multiplication, to obtain OF, making it theoretically superior to the gradient-based method.

### Funding
The authors received no funding for this work.

### Competing Interests
The authors declare there are no competing interests.

### Author Contributions
- Ting-Wei Shen conceived and designed the experiments, performed the experiments, analyzed the data, performed the computation work, prepared figures and/or tables, authored or reviewed drafts of the article, and approved the final draft.
- Mao-Hsiu Hsu conceived and designed the experiments, prepared figures and/or tables, and approved the final draft.
- Chun-Hsu Shen conceived and designed the experiments, analyzed the data, prepared figures and/or tables, and approved the final draft.
- Wen-Fang Wu conceived and designed the experiments, authored or reviewed drafts of the article, and approved the final draft.
- Yu-Chiao Lu performed the experiments, authored or reviewed drafts of the article, and approved the final draft.
- Chia-Chun Chu performed the experiments, authored or reviewed drafts of the article, and approved the final draft.

### Data Availability
   The raw data and code are available in the Supplemental Files.

### Supplemental Information
Supplemental information for this article can be found online at http://dx.doi.org/10.7717/peerj-cs.1342#supplemental-information.

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
