# Peer review of "An effective fingerprint orientation field estimation method using differential values of grayscale intensity"

_PeerJ Computer Science, doi:10.7717/peerj-cs.1342_

## Round 0.1 · original submission · Major Revisions

Revise as per the reviewer comments.

Reviewer 1 ·

Basic reporting

no comment

Experimental design

no comment

Validity of the findings

Found the lack of the enough raw data in experiment to make the concrete conclusion.

Additional comments

Comparison of the proposed methodology to improve the fingerprint estimation is just compared with one classic gradient-based method and not mentioning any other modern algorithms.

Reviewer 2 ·

Basic reporting

1) Most of the references were published before 2015 and cannot show the novelty of the article. It is recommended to review the latest research in the field and illustrate the innovation of the proposed method.

Experimental design

no comment

Validity of the findings

1) As stated in the title, Figure. 9 shows "the performance of the gradients-based OF estimation and the proposed OF estimation compared on 1000 fingerprints blocks". But only the results of the proposed method are listed in the bar chart. Please check.
2) As shown in the two experiments, the performance of the proposed method is similar to that of the conventional method. Additional experiments are suggested to better illustrate the superiority of the proposed method.
3) It is suggested to further explain why the proposed method is theoretically superior to the existing methods.

Additional comments

In this paper, a new fingerprint orientation field estimation method is proposed based on the differential values of grayscale intensity. Overall, the experimental settings and language expression is clear. But the experimental data and analysis are relatively insufficient to support the novelty of the proposed method. Additional literature review and experiments are suggested to enhance the article.

·

Basic reporting

The authors claim an innovative method to pre-process fingerprint images over the SOTA. There are many other issues with the basic reporting, as below:
1. Insufficient background and context. For e.g,
(i) Why does the Gabor filter require orientation field (OF) information and how does it estimate the OF in current practice ? What is the need for the current method?

2. The standard of some figures requires attention. For e.g,
(a) Figures 7 and 8 seem to be related, hence may be presented as such.
(b) Figure 9. There is no comparison being made - caption and the figure do not tally.
(c) Figure 10 appears redundant. The information presented in the figure is already there in the text. Nothing new is being conveyed.

3. Some Raw data has been shared, but it is not clear what it represents. Further LabVIEW code has been shared, but it is baffling why the authors have chosen this framework to code the analysis.

4. The authors have used Gaussian noise to generate blurred images. But no specifics regarding the Gaussian blurring has been provided

5. There are many issues with the language, and language editing is necessary. For e.g,
(i) lines 66-67 "In contrast, if the straight lines are closer to the orientations of fingerprints, the
grayscale value would change less sensitively." -- could be improved
(ii) lines 75-76 "In our experimental result, the performance of the proposed OF estimation is along with the number of orientations." -- could be improved
(iii) lines 102-103. etc

Experimental design

(i) The research question has not been motivated, or motivated adequately. The authors merely state (lines 73-74) "This study is based on this concept to develop a simple and reliable OF estimation algorithm by replacing differential operations with convolution calculations to simplify the calculation."

(ii) lines 94-95: ...the result of the convolution calculation is related to the differential of the grayscale value in each orientation.
-- this logic in this assertion needs to be clarified explicitly in the interest of an interdisciplinary audience.

(iii) lines 128-129: The OF estimated by the proposed OF estimation algorithm is similar to that of the classic gradient-based method.
However the authors have provided no performance measures for the gradient-based method. With no supporting data provided to this statement, the authors may not be doing the readers any favor in understanding their work. The authors need to pay attention to bolster their findings appropriately.

(iv) Significance of the findings has not been estimated, and this is a key concern with the work. Without significance studies, the manuscript may not suitable for publication in PeerJ.

Validity of the findings

1. In lines 129-130, the authors state, '23 blocks with a deviation of more than 20 degrees were
observed.'
Are these in the periphery (edges) or in the centre of the image? This makes a difference. If in the centre, then the image integrity could be jeopardized. Alternatively if in the periphery then the author's conclusion could be valid.
The reporting in the paper leaves a lot of room for improvement.

2. Further in lines 135-136, authors mention, 'In addition, error blocks occur most often in the core
area in fingerprint images, and identifying parallel fingerprints is difficult in the core area.'
This seems a serious limitation of the proposed method. The core area is probably the key area in the image, and failure to perform in the core area (or showing much error in this area) is likely to reduce confidence in the method. The method does not fulfil its main objective apparently.

Additional comments

In summary, this reviewer feels that any merit in the proposed method needs to be brought out with a larger datasets, more thorough analyses etc. This would enable establishing the validity of the method, rather than the use of ad hoc reliability assessments as currently described.

---

## Round 0.2 · Major Revisions

Revise as per reviewer comments.

Reviewer 2 ·

Basic reporting

no comment

Experimental design

no comment

Validity of the findings

no comment

Additional comments

The comments have been well addressed.

·

Basic reporting

The manuscript English has also improved, following professional language editing.
The authors have also clarified certain areas in their manuscript in the interest of an interdisciplinary audience.
In response to this reviewer's request, the authors have updated the raw data and made it more useful. They have also increased the size of the dataset apparently, though it is not mentioned by exactly how much.

Experimental design

The authors have addressed most of the comments. Whether the performance of different techniques on the same dataset are significantly different or not could be assessed using statistical testing.

Validity of the findings

In response to my comment, "Further in lines 135-136, authors mention, 'In addition, error blocks occur most often in the core area in fingerprint images, and identifying parallel fingerprints is difficult in the core area.' This seems a serious limitation of the proposed method. The core area is probably the key area in the image, and failure to perform in the core area (or showing much error in this area) is likely to reduce confidence in the method. The method does not fulfil its main objective apparently.", the authors state that,
"The core area is like a semi-circle or an arc, therefore it is difficult to define the OF in core area of fingerprint. The circumstance of OF estimation in core area in the classic gradient-based method is similar. (lines 162-164)"
However this does not address the concern that was raised, merely says that the limitation exists in another method too. It will be signifcant to quantitatively estimate out the impact of this limitation on the performance measures.

---

## Round 0.3 · accepted · Accept

It can be accepted now. It is improved according to the reviewers' comments.

·

Basic reporting

no comment

Experimental design

no comment

Validity of the findings

The authors have accepted the suggestions and carried out revisions or responded to the concerns suitably.